# Extracts of Waste from Poplar Wood Processing Alleviate Experimental Dextran Sulfate-Induced Colitis by Ameliorating Oxidative Stress, Inhibiting the Th1/Th17 Response and Inducing Apoptosis in Inflammatory Lymphocytes

**DOI:** 10.3390/antiox10111684

**Published:** 2021-10-26

**Authors:** Wenjie Wang, Yiwei Zhang, Jiamin Cao, Jiahui Xu, Linguo Zhao, Xianying Fang

**Affiliations:** 1College of Chemical Engineering, Nanjing Forestry University, Nanjing 210037, China; wangwenjie@njfu.edu.cn (W.W.); zhang_yiwei0103@wuxiapptec.com (Y.Z.); cjm@njfu.edu.cn (J.C.); xujiahui@njfu.edu.cn (J.X.); 2Co-Innovation Center for Sustainable Forestry in Southern China, Nanjing Forestry University, Nanjing 210037, China; 3Jiangsu Co-Innovation Center of Efficient Processing and Utilization of Forest Resources, Nanjing Forestry University, Nanjing 210037, China

**Keywords:** poplar, feed additives, wood processing wastes, colitis, inflammation, Th1/Th17

## Abstract

As a fast-growing tree, poplar is widely planted and typically used for wood processing in China. During poplar wood processing, a large amount of poplar sawdust (PS) and poplar leaves (PL) are produced and abandoned. To make full use of poplar resources and clarify the use of poplar as a feed additive, the active ingredients in PS and PL were extracted and isolated, and the anti-inflammatory effects of the extracts on mice with dextran sulfate sodium (DSS)-induced colitis were investigated. In vitro anti-inflammatory experiments showed that the ethyl acetate extract of PS and PL (PSE and PLE, respectively) could significantly inhibit the proliferation of concanavalin A (Con A)-activated lymphocytes. Salicortin, tremulacin and salireposide were identified in both PSE and PLE. Oral administration of PSE and PLE rescued DSS-induced colonic shortening, repaired tissue damage, and decreased the disease activity index (DAI). The antioxidant capacity, including the increased activities of glutathione peroxidase (GSH-Px), total superoxide dismutase (T-SOD and catalase (CAT) and decreased activity of myeloperoxidase (MPO), in the colons of mice with colitis was enhanced through the activation of ERK after treatment with PSE and PLE. The ratio of Th1 to Th17 cells, which can lead to inflammation in the spleen, was significantly decreased by the administration of PSE and PLE, while the phosphorylation of related transcription factors (p65, STAT1, and STAT3) was inhibited. Furthermore, PSE and PLE could induce apoptosis in Con A-activated lymphocytes, which may be associated with the increase in p-TBK1, as the molecular docking results also indicated that salireposide in PSE and PLE could interact with the TBK1 protein. Overall, our study provides a promising feed additive for improving intestinal inflammation in animals and a method for the full utilization of poplar resources.

## 1. Introduction

Inflammatory bowel disease (IBD) is a chronic multifactorial inflammatory disease affecting the gastrointestinal tract. Both humans and animals can suffer from IBD [1]. It is triggered by excessive pro-inflammatory cytokine production, sustained T-cell activation and oxidative stress-induced epithelial mucosal cell death [2]. Patients with colitis often present with weight loss, internal bleeding, diarrhea, and extensive colonic mucosal and submucosal lesions [3,4]. Most standard treatments for IBD use pharmaceutical agents, such as aminosalicylates, immunosuppressants, biologics, or combination therapies proved to have certain harmful side effects [5]. In order to develop new and accessible drugs for IBD, especially for the application in animals, much attention has been focused on plant extracts [6]. Usually, plants are rich in polyphenols, which show good antioxidant activity and can modulate the immune response in the intestinal system [7,8]. Generally speaking, plant extracts may be a more effective strategy for improving health.

Poplar is widely distributed in eastern China and is one of the most important tree species for afforestation and wood processing. The range of uses of poplar plantations is expanding, with applications in plywood, pulp and paper, solid wood plywood, and particle board [9]. The expansion of the poplar processing industry is accompanied by an increased amount of abandoned poplar sawdust (PS) and poplar leaves (PL). The direct disposal of PS and PL will not only cause a waste of resources, but also increase the burden of the environment. Some farmers like to use the bark and leaves of poplar trees as nutrients to feed poultry. However, the use of poplar, especially PS and PL, in animal feed is worthy of further study. The bark and leaves of poplar are rich in phenolic glycosides, flavonoids, organic acids, and terpenoids, which play an important role in its anti-inflammatory [10], antioxidant [11], and antibacterial [12] properties. As a series of known glycosylated and esterified derivatives of salicylic alcohols, popcinol glycosides have good biological activity. Salireposide, a component of herbal combination, protects mice from abdominal sepsis and death caused by E. coli and modulates the immune response [13]. The ethyl acetate extract of poplar bark has been reported to have the ability to scavenge ABTS free radicals [14]. It would be meaningful if the poplar industrial wastes can be applied in feed additive.

Isolating monomers from the by-products of processing is cumbersome and often of limited effectiveness, and crude extracts could exert a wider range of pharmacological activities [15]. More researches were focused on using rough extraction methods to obtain the maximum utilization value [16]. However, different extraction methods will have a certain impact on the activity of the extract, and screening out the most appropriate extraction method is meaningful [17]. Although many studies have demonstrated the anti-inflammatory effects of compounds in poplar [18,19], the application prospect of PS and PL extracts in animal intestinal health is uncertain. In order to find the application value of poplar processing waste, the anti-inflammatory and antioxidant activities, especially the improvement on DSS-induced colitis in mice, of PS and PL extracts were explored, and the underlining mechanisms were also investigated in this study.

## 2. Materials and Methods

### 2.1. Materials

DSS (molecular weight: 40,000) was purchased from Aiaddin Biochemical Technology Co. (Shanghai, China). Dimethyl sulfoxide (DMSO), 5-ASA, methyl thiazolyl tetrazolium (MTT), BCA protein, Concanavalin A (Con A), were purchased from Sigma-Aldrich (St. Louis, MO, USA). Roswell Park Memorial Institute-1640 (RPMI-1640) medium (10% FBS, 1% penicillin and streptomycin double antibody) was purchased from Gibco (Massachusetts, USA).

Total superoxide dismutase (T-SOD), catalase (CAT) and glutathione peroxidase (GSH-Px) colorimetric kits were purchased from Nanjing Jiancheng Institute of Biological Engineering (Nanjing, China). Antibodies to phosphorylated STAT1, phosphorylated STAT3, phosphorylated p65, phosphorylated TBK1, cleaved Caspase-3, Caspase3, β-actin were purchased from Cell Signal Technology (Beverly, MA, USA). Antibodies to STAT1, STAT3, p65 were purchased from Phorbol myristate acetate (PMA, Hangzhou, China), Brefidomectin A (BFA, Hangzhou, China), FIX&PERM Kit, FITC-anti-CD3, PerCP-Cy5.5-anti-CD4, PE-Cy7- anti-IFN-γ, PE-anti-IL-17A, FITC-anti-Annexin V were purchased from Doko (UNIQUE) Biotechnology Co. (Hangzhou, China).

### 2.2. Animals

Female BALB/c mice (7 to 8 weeks old, weighing 18–22 g) were provided by the Experimental Animal Centre of Yangzhou University (Jiangsu, China). Animals were fed a standard rodent diet, had free access to water and were housed in a room at 23 ± 2 °C with a 12-h light/dark cycle according to international recommendations. Animal welfare and experimental procedures followed the Guide for the Care and Use of Laboratory Animals (National Institutes of Health, USA). The present study was conducted in accordance with the Animal Care and Protection Committee of Gulou Hospital, Nanjing University (SYXK 2004-0013) and the related ethical regulations of our university. The authors confirmed that all animals were provided with humane care and that all animal experiments were performed in accordance with relevant guidelines and regulations. All authors complied with the ARRIVE guidelines for experiments.

### 2.3. Preparation and Analysis of the Extract of Poplar Sawdust (PS)and Poplar Leaves (PL)

Poplar sawdust and leaves were purchased from Fu Qing Wood Processing Plant, Jiangsu, China, and authenticated as originating from *Populus euramericana* on the morphological characteristics by the International Cultivar Registration Center (Nanjing Forestry University). The extraction method was based on the method of Zhang Xinfeng et al. [13]; PS and PL powder were extracted by 90% ethanol, respectively, three times at 60 °C under reflux of 2 h. The extracts were filtered (Qualitative Filter Paper, Beyotime Biotechnology, Shanghai, China) and then concentrated under reduced pressure until no alcohol taste. A portion of the ethanol extract was freeze-dried and collected. The other parts were dissolved in water and then extracted with petroleum ether and ethyl acetate sequentially, each fraction was collected and dried by evaporation under reduced pressure. The total ethanol extract, the petroleum ether extract, and the ethyl acetate extract were referred to as EE, PE, and EA of PS and PL in this paper. For activity analysis experiments, the extracts of PS and PL were dissolved in DMSO at a concentration of 100 mg/mL as stock solution and tested at concentrations diluted with medium before each experiment.

The components in the extracts of PS and PL were analyzed by liquid chromatography-mass spectrometry system (LC-MS). Separation was performed on a C18 column (250 mm × 4.6 mm, 5 μm, S. No. USAG008115, Santa Clara, CA, USA) at 30 °C with a flow rate of 0.5 mL/min. gradient elution with a mobile phase consisting of 0.1% formic acid in water (A) and Anhydrous acetonitrile(HPLC/Spectro) (B): 0–5 min, 95–95% A (*v*/*v*); 5–20 min, 95–87% A (*v*/*v*); 20–30 min, 87–60% A (*v*/*v*).

### 2.4. Cell Viability and Apoptosis Assays

Lymphocytes isolated from female BALB/c mice were resuspended in complete RPMI 1640. They were cultured in 96-well plates (5 × 10^6^ cells/mL, *n* = 3) and cultured at 37 °C under humidified conditions with 5% CO_2_. In the proliferation assay, lymphocytes were stimulated with 4 μg/mL Con A and/or no PS/PL extracts (EE, PE, EA) for 24 h. 20 μL (4 mg/mL) MTT was then added to each well 4 h before the end of the experiment. After 4 h, cells were collected and supernatant was removed. 200 μL DMSO was then added to each well and mixed for 10 min. The optical density was read at 560 nm using an ELISA reader(Molecular devices, Silicon Valley, CA, USA).

For the apoptosis assay, lymphocytes were cultured on 6-well plates and incubated at 37 °C with 5% CO_2_ humidification. Cells were stimulated with 4 μg/mL Con A and/or no PSE/PLE (the ethyl acetate extracts of PS and PL). After 24 h, 10 μL PI and 5 μL Annexin V-FITC were added and incubated for 15 min, then collected and resuspended with Binding Buffer. Apoptosis was calculated using the BD FACS Canto II Flow cytometer (Becton, Dickinson and Company, Franklin Lakes, NJ, USA) and analyzed by FlowJo 10 (Becton, Dickinson and Company, NJ, USA).

### 2.5. Induction and Assessment of Colitis

For each experiment, mice were divided into 7 experimental groups (*n* = 8 per group). Except for the normal group, mice were challenged with 3.5% (wt/vol) DSS dissolved in drinking water for 7 days and then given normal water for 3 days until sacrificed. PSE, PLE (20, 40 mg/kg), or 5-ASA (300 mg/kg) were dissolved in phosphate buffered saline (PBS) solution and administered orally daily from day 3 to day 10. The normal group was kept as the vehicle-treated control and treated with saline in the same route as PSE. Mice were observed for weight loss, fecal consistency and defecation. The Disease Activity Index (DAI) is calculated as the sum of the following three parameters: (a) weight loss (0 = no loss, 1 = 0–5% loss, 2 = 5–10% loss. 3 = 10–15% loss, 4 = more than 15% loss); (b) fecal consistency (0 = normal, 1 = mildly soft, 2 = soft and wet, 3 = semi-dilute stool, 4 = dilute stool); (c) fecal bleeding (0 points = normal, 2 points = slight bleeding, 4 points = severe bleeding). After 10 days of induction, mice were sacrificed. The whole colon was quickly removed and washed with PBS and the length of the colon was measured.

### 2.6. Analysis of Helper T Cell (Th) Subsets in Splenic Lymphocytes

Groups of splenic lymphocytes were collected on day 10 of induction, activated with 1 μL Phorbol myristate acetate (PMA)/Ionomycin Mixture (250×) and 1 μL Brefidomectin A (BFA)/monensin Mixture (250×) for 6 h at 37 °C and stained with anti-mouse CD3, FITC and anti-mouse CD4, PerCP-Cy5. Cells were incubated in the dark for 15 min to be fixed and permeated, then were washed twice and stained with anti-mouse IFN-γ, PE-Cy7 and anti-mouse IL-17A, PE for 15 min. After two washes, cells were resuspended in 500 μL flow cytometry staining buffer and samples were analyzed by flow cytometry.

### 2.7. Quantitative RT-PCR

Total RNA was extracted from the mouse spleen and then reverse transcribed to cDNA using a kit. Quantitative PCR was performed with the Step One Plus Real-Time PCR System using TB Green *Premix Ex Taq* II (Takara Bio Inc., Beijing, China). PCR cycling conditions were as follows: 1 cycle of 95 °C for 30 s, followed by 40 cycles of 95 °C for 5 s, and 60 °C for 30 s. RNA expression was quantified relatively using β-actin as a control and analyzed using the delta-delta Ct (2^−ΔΔCT^) method. The primer sequences were as shown in Table 1.

### 2.8. Western Blot Analysis

Proteins were quantified using the Total Tissue Protein Extraction Kit (Beyotime Biotechnology, Shanghai, China). Proteins lysed from lymphocytes and animal tissues were separated by SDS-PAGE and transferred to polyvinylidene difluoride (PVDF) membranes. Membranes were blocked with 5% skimmed milk and incubated with antibodies overnight at 4 °C, followed by 2 h incubation with HRP-coupled secondary antibodies. Protein bands were detected with ECL detection reagents. The intensity of the bands was quantified using Image J software, β-actin was used as an internal control.

### 2.9. Histological Analysis

The distal portion of the colon was fixed in formalin, embedded in paraffin, sectioned at 4 mm thickness, and stained with hematoxylin and eosin (H&E) for histological analysis. Histological damage and inflammation were observed under the microscope using a light microscope. Histological assessment (*n* = 8) was performed in a blinded manner, using a modified validated scoring system previously described [20].

### 2.10. Molecular Docking Analysis

The three-dimensional crystal structure of TANK-binding kinase 1(TBK1) (PDB ID:6CQ0) was retrieved from the RCSB Protein Data Bank (https://www1.rcsb.org/structure/6CQ0, accessed on 20 May 2021). The downloaded protein was further optimized to remove unwanted protein subunits, het atoms and water molecules, add missing heavy atoms and formal charges by Schrödinger’s protein preparation wizard (Maestro 10.1, Schrödinger, NY, USA). Chain A of TBK1 protein was taken for further preparation. The salireposide ligands in SDF format were downloaded from Pubchem and exported in PDB format. Then the ligands were prepared using the LigPrep module of the Schrödinger software suite so that the salireposide was optimized with suitable parameters like using OPLS3 force field, 2D to 3D conversion, generation of all possible ionization states at pH 7.0. Schrödinger Glide (version 2015, Schrödinger, NY, USA) was used for docking. The ligand interaction diagram was exported after confirming the optimal structure.

### 2.11. Statistical Analysis

All results are expressed as the mean ± SD. Student’s *t*-test and the one-way ANOVA test were used for statistical analyses of the data. All statistical analyses were conducted using SPSS 10.0 statistical software (SPSS, Chicago, IL, USA). Cases in which *p* values of <0.05 were considered statistically significant.

## 3. Results

### 3.1. The Ethyl Acetate Extracts from the Ethanol Extract of PS and PL Showed Excellent Anti-Inflammatory Effects

EE, PE, and EA were used to compare anti-inflammatory effects. An in vitro Con A-activated lymphocyte proliferation model was used to initially determine the differences in activity between these three extracts. The extracts were added and incubated for 24 h, and MTT was used to examine cell viability. As shown in Figure 1A, the proliferation of lymphocytes was more inhibited in the EA group than in the other groups. Therefore, ethyl acetate extracts of the ethanol extracts of PS and PL were used for animal experiments and were renamed PSE and PLE, respectively. Then, a preliminary analysis of the composition of PSE and PLE were performed by LC-MS. As shown in Figure 1B, compared with the reported research studies on poplar and combined with relevant references [13,21], major phenolic glycosides in poplar, including salicortin, tremulacin, chrysin, and salireposide, were identified.

### 3.2. PSE and PLE Ameliorate the Pathological Changes of DSS-Induced Colitis in Mice

In this study, a DSS-induced colitis mouse model was used to evaluate the therapeutic effects of PSE and PLE. Mice were administered 3.5% DSS in drinking water to induce inflammation in vivo. Compared with the control group, the weight of model group mice decreased significantly from the fifth day and the gap reached the maximum on day 10. The DAI score increased significantly after DSS intake, on the sixth day mice in the model group showed perianal massive hemorrhage. Moreover, the DSS group exhibited severe colitis symptoms evidenced by the disappearance of goblet cells, superficial epithelial cells, and the increase of inflammatory cell infiltration in colon tissues. These phenomena illustrate the success of modeling. As shown in Figure 2, mice treated with PSE and PLE showed improved weight loss (Figure 2A) and significant reductions in the DAI (Figure 2B). 5-aminosalicylic acid (5-ASA), which has been widely used to treat IBD in clinical settings, may play a therapeutic role by reducing the expression levels of tumor necrosis factor alpha (TNF-α) in colon tissues. In our experiments, 5-ASA was evaluated as a positive control for the treatment of IBD.

Shortening of the colon is considered to be an important indicator of the severity of colitis in mice. Colonic shortening caused by inflammation was alleviated by PSE and PLE to varying degrees. In addition, better amelioration of colonic shortening was realized by high doses (40 mg/kg) of both extracts (Figure 2C,D).

Colitis is characterized by the infiltration of inflammatory cells and damage to colonic tissue. Severe pathological changes were observed in colonic samples from DSS-induced mice (Figure 2E): inflammatory cells infiltrated the colon, and tissue was destroyed. Mice treated with PSE and PLE were protected against severe histological damage, and higher doses of the extracts showed better therapeutic effects, with the PSE and PLE groups showing mean histological scores of 2.13 and 2.25, respectively, in response to 40 mg/kg, while the DSS group showed a high score of 3.63.

### 3.3. PSE and PLE Inhibit Oxidative Stress in the Colon of DSS-Induced Colitis Mice

Oxidative stress is an important factor that contributes to the destruction of colonic mucosa and exacerbating colitis. Inhibiting oxidative stress can effectively improve colitis [22]. In the colitis model group, antioxidant enzyme activities in the colon were inhibited, and the enzymatic activities of GSH-Px, CAT and T-SOD, which can enhance antioxidant function, were significantly decreased compared to those in the Ctrl group (Figure 3A–C). MPO, which is a promoter of oxidative stress, reflects the degree of inflammatory cell infiltration in the colon to some extent. The MPO level was significantly higher in DSS-induced colitis mice, and this effect was inhibited by PSE and PLE (Figure 3D).

### 3.4. PSE and PLE Enhance Antioxidant Capacity in Mice through the Activation of ERK/Nrf2 Pathway

RNA was extracted from colonic tissue from each group of mice and quantified by RT–PCR to further investigate the roles of PS and PL in enhancing the antioxidant capacity. The results showed that the RNA levels of *sod* and *gsh-px* were increased by PSE and PLE, and this effect could promote the production of antioxidant enzymes (Figure 4A).

Activation of the ERK/Nrf2 pathway has been shown to regulate the antioxidant capacity of cells [23,24] and prevent damage caused by oxidative stress. The PCR results showed that the expression of *nrf2* and its downstream gene *ho-1* were activated in PSE- and PLE-treated mice, and high doses of PSE and PLE promoted the expression of these genes (Figure 4A). Western blot analysis revealed that the phosphorylation level of ERK was significantly increased compared to that in the model group (Figure 4B). This result demonstrates that PSE and PLE may indeed enhance the antioxidant capacity of mice by activating the ERK/Nrf2 pathway.

### 3.5. PSE and PLE Regulate Cytokine Profiles in the Spleen of Colitis Mice

In previous studies, an imbalance in T cell subsets has been identified as one of the key triggers of IBD. Th1 and Th17 cells are closely associated with IBD, as these cells can produce proinflammatory cytokines (e.g., IFN-γ, TNF-α and IL-17A), which can lead to inflammation [22,25]. On day 10 after induction, fresh spleens were used for flow cytometry analysis of cells. In our study, DSS-induced mice showed an increased ratio of Th1 to Th17 cells. In contrast, mice treated with PSE and PLE had reduced ratios (Figure 5A,B). Total RNA was isolated from spleens, and the RNA expression of cytokines was analyzed by quantitative RT–PCR. The results showed that proinflammatory cytokines, including tumor necrosis factor-α (*tnf-α*), *ifn-γ*, and *il-17A,* were overexpressed in the spleens of DSS-induced mice and were suppressed by PSE and PLE (Figure 3C).

### 3.6. PSE and PLE Down-Regulate the Phosphorylation of STAT1, STAT3 and p65 in the Model of Inflammation

Activation of transcription 1 (STAT1) and transcription 3 (STAT3) signaling is essential for the differentiation of Th1 and Th17 cells and the secretion of inflammatory cytokines [26]. The expression level of nuclear factor kappa B (NF-κB), which is one of the main components mediating inflammation, is equally important in IBD [26]. Studies on the changes in protein expression levels would be helpful for us to understand the pharmacological mechanism of PSE and PLE in colitis mice. Western blot analysis showed that phosphorylation of STAT1, STAT3, and p-65 were inhibited by PSE and PLE compared to that in the model group (Figure 6A), suggesting that PSE and PLE can improve inflammation through the STAT1/STAT3 and NF-κB signaling pathway.

Next, a Con A-stimulated lymphocyte model was established to verify the anti-inflammatory activity of PSE and PLE. As shown in Figure 6B, after 24 h of Con A stimulation, the phosphorylation levels of STAT1, STAT3, and p-65 in the presence of PSE and PLE were apparently decreased compared to those in the model group. Similarly, the levels of proinflammatory cytokines were analyzed by quantitative RT-PCR. The results revealed that proinflammatory cytokines, including tumor necrosis factor-α (TNF-α), IFN-γ, and IL-17A, were overexpressed in an in vitro Con A-stimulated lymphocyte model and were reversed by the addition of PSE and PLE (Figure 6C).

### 3.7. PSE and PLE Inhibit Proliferation of Activated Lymphocytes through Induction of Apoptosis

The proliferation of lymphocytes activated by Con A was shown to be robustly inhibited by PSE and PLE in a previous study, but the mechanism that PSE and PLE inhibit lymphocyte proliferation and, thus, improve inflammation is unclear. Figure 7A,B shows that after 24 h of stimulation by Con A, lymphocytes were cultured with PSE and PLE, which promoted late apoptosis within 24 h compared to that in the model group, while early apoptosis was not affected. The in vitro results suggest that the induction of apoptosis contributes to PSE- and PLE-mediated inhibition of lymphocyte-mediated inflammation.

### 3.8. PSE and PLE May Induce Lymphocyte Apoptosis and Ameliorate Inflammation through Activation of STING-TBK1 Pathway

A previous study showed that a lack of STING signaling could not limit STAT3, NF-κb pathway activation [27,28], thereby exacerbating colitis and inducing colon cancer in colitis mice. In B lymphocytes, STING has been reported to have mechanisms that negatively regulate the JAK-STAT1 signaling pathway [29], and these findings facilitate our interpretation of the pathways through which PSE and PLE may inhibit the STAT 1, STAT 3 and NF-κB pathways. Furthermore, the activation of STING triggers a cascade of responses that support an antiproliferative cellular state, including cellular senescence and ultimately cell death [30], and activation of STING specifically promotes apoptosis of T cells and B cells [31,32,33], a reduction in inflammation would be beneficial by inducing apoptosis in activated lymphocytes.

In our experiments, TBK1 phosphorylation levels were reduced in the spleens of mice with colitis, and the STING-TBK1 signaling pathway was inhibited, whereas the STING-TBK1 signaling pathway was activated and lymphocyte apoptosis was induced by PSE and PLE (Figure 8A). Then, a Con A-induced lymphocyte proliferation model was used to verify our hypothesis. Cells were divided into a Con A-stimulated model group and PSE and PLE addition groups. The expression of related proteins was measured at 1 h, 2 h and 4 h. As shown in Figure 8B, the phosphorylation of TBK1 was significantly increased, and NF-κB activation was inhibited by the addition of PSE and PLE. At 4 h, the protein level of cleaved Caspase3 was higher in the PSE and PLE groups than in the model group.

Next, molecular docking analysis was used to aid in verification. Salireposide (a major compound in our extract) was used to demonstrate an interaction with the TBK1 protein. There was a strong interaction between TBK1 and salireposide, and salireposide could insert into the active domain of TBK1 kinase and interact through hydrogen bonds and π-π interactions (Figure 8C). The docking results showed significant π-π conjugation between the loops of CYS89, MET142, ALA36, VAL23, and TBK1. In addition, salireposide formed three hydrogen bonds with ALA21, ASP157, and LYS38 of TBK1, conferring stability to the docked conformation (Figure 8D). These results suggest that PSE and PLE could treat colitis through the STING-TBK1 pathway.

## 4. Discussion

Studies have shown that the long-term use of antibiotics can lead to the destruction of the intestinal flora in animals, thereby inducing more severe colitis [34]. Poplar waste is a promising material which can be used for the production of alternatives to antibiotics to promote intestinal health. However, the therapeutic effect and mechanism of poplar on colitis have not been systematically studied. Our initial results show that the proliferation of Con A-activated lymphocytes can be inhibited by PSE and PLE. Therefore, the active ingredients in poplar waste may be used to treat inflammation-related health problems. Intestinal inflammation affects the healthy growth of animals and may lead to death if conditions worsen, and the related issues are a large challenge for the animal farming industry. Furthermore, excessive inflammatory activation exacerbates oxidative stress, which in turn leads to the dysregulation of endogenous antioxidants, resulting in damage to the function of the tight junctions of normal colonic epithelial cells and epithelial cell integrity. Regulating the levels of antioxidants and oxidants and enhancing antioxidant capacity are also key in treating colitis in animals. In this study, PSE and PLE enhanced the activity of antioxidant enzymes (Gsh-px, SOD and CAT) in vivo and protected mice from further colon damage.

Excessive proliferation of helper T cells and the release of proinflammatory cytokines are key factors in the development of colitis. Recent studies have shown that immune homeostasis in the gut depends on the regulation and balance of CD4^+^ T cells. The activation of Th1 and Th17 cells can lead to intestinal inflammation, such as IBD, and inhibiting the Th1/Th17 response can ameliorate inflammation in IBD [35,36,37]. DSS-induced colitis exhibits a predominance of Th1/Th17-type cytokines, which are widely used to model IBD in patients [25]. With this model, the function of Th1/Th17 cells and the gene expression of Th1/Th17-related cytokines, including IFN-γ, TNF-α and IL-17A, were inhibited by the administration of PSE and PLE. Moreover, PSE and PLE suppressed the activation of STAT 1/STAT 3, which is a key transcription factor in the Th1/Th17 response [26]. The phosphorylation of NF-κB and its nuclear translocation can cause severe intestinal inflammation [38]. However, the activation of NF-κB in models was sufficiently attenuated by PSE and PLE. Thus, PSE and PLE may be good options for the treatment of colitis.

Aberrant or excessive production of proinflammatory cytokines is associated with uncontrolled local and systemic inflammation, leading to cell death and often irreversible tissue damage. The inhibition of excessive effector functions is a key factor in the treatment of inflammatory diseases [39,40]. In IBD animals, the infiltration of inflammatory cells damages colonic tissues and is detrimental to the lives of these animals. In this study, the mechanism by which PSE and PLE inhibit lymphocytes was explored. Flow cytometry showed that PSE induced apoptosis, which is effective in treating colitis. This finding also explains the increased levels of c-Caspase 3 in the spleens of colitis mice, suggesting that the induction of apoptosis by PSE and PLE is essential for inhibiting lymphocyte proliferation and activation. 

Elevated expression of the STING signaling pathway in immune cells has been shown to specifically promote lymphocyte regulation, and overstimulation of the STING/TBK1 pathway triggers apoptosis in lymphocytes [31,33]. Moreover, studies have shown that the cytosolic sensor STING is important in maintaining gut homeostasis and preventing autoimmune conditions, such as colitis [41]. Modulating the STING/TBK1 pathway to induce inflammatory cell apoptosis can be an effective means of treating colitis. Interestingly, in this study, mice treated with PSE and PLE had activated p-TBK1 and c-Caspase3 compared to those in the model group, suggesting that PSE and PLE may induce inflammatory lymphocyte apoptosis by modulating this pathway in animals. Similarly, Con A-stimulated cells treated with PSE and PLE showed increased levels of phosphorylated TBK1 compared to those in the model group. Molecular docking analysis showed that the compounds in PSE and PLE could interact with TBK1. These data proved that lymphocyte apoptosis induced by PSE and PLE may be related to the STING/TBK1 pathway. This finding would help us to better explain the function of PSE and PLE as feed additives.

## 5. Conclusions

In conclusion, our results show that PSE and PLE have good anti-inflammatory and antioxidant activities, especially in the treatment of DSS-induced colitis. Animal weight loss, DAI scores, and colon tissue damage in colitis mice were significantly improved by PSE and PLE. Further study indicated that the oxidative stress and inflammation were also significantly inhibited. The regulatory mechanism of PSE and PLE in tissue is mainly related to the ERK, STAT, and signaling pathways. Moreover, salireposide in PSE and PLE may be related to the activation of TBK1, which gives them the ability to induce lymphocyte apoptosis. As a fast-growing tree, poplar can not only be used for wood processing, but also has the potential to be used as a healthy feed additive by utilizing its processing wastes. It is critical to develop the animal farming industry under the conditions of reduced land and increased people, because of poor living environments and inefficient utilization of plant resources. 

## Figures and Tables

**Figure 1 antioxidants-10-01684-f001:**
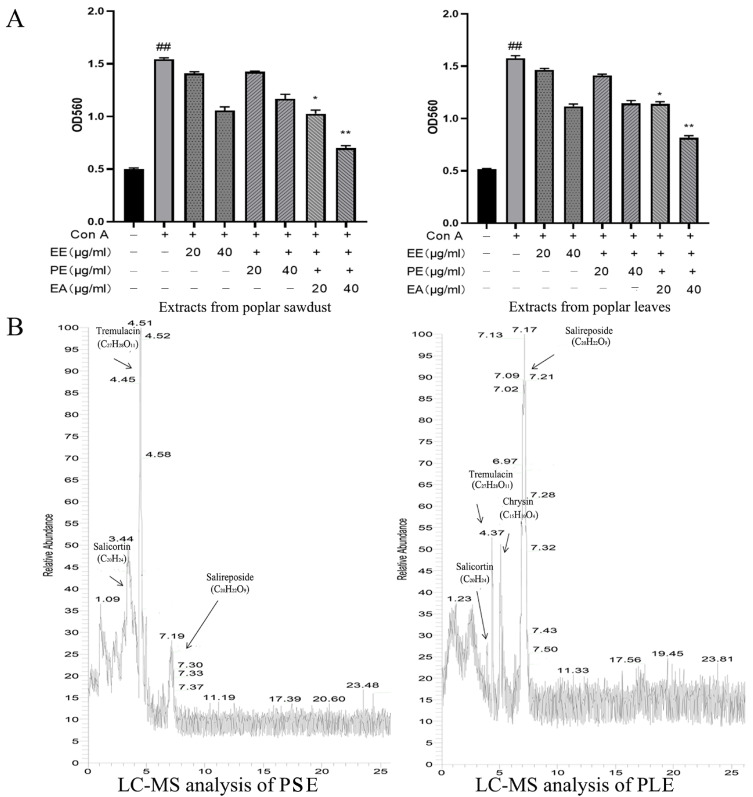
Extracts of poplar waste showed anti-inflammatory activity. (**A**) Three fractions were obtained from poplar sawdust and poplar leaves. Poplar sawdust (PS) and poplar leaves (PL) were extracted by ethanol and then extracted by petroleum ether and ethyl acetate. Extracts were divided into three groups: total ethanol extract (EE), petroleum ether extracted phase (PE) and ethyl acetate extracted phase (EA). The inhibition rates on the proliferation of Con A-activated lymphocytes were analyzed, both EA showed excellent anti-inflammation activity. Ethyl acetate extracted phase of PS was named PSE and ethyl acetate extracted phase of PL was named PLE. The inhibition rate of PSE was 80.6 ± 2.35%, PLE was 69.43 ± 1.4%. (**B**) LC-MS analysis of the components in the poplar extracts of PS and PL. Data are represented as mean ± SD, of three experiments *n* = 3. The “+” indicates that the substance was added, and the “–” indicates that the substance was not added. ^##^
*p* < 0.01 vs. blank (Con A negative), * *p* < 0.05, ** *p* < 0.01 vs. Con A control.

**Figure 2 antioxidants-10-01684-f002:**
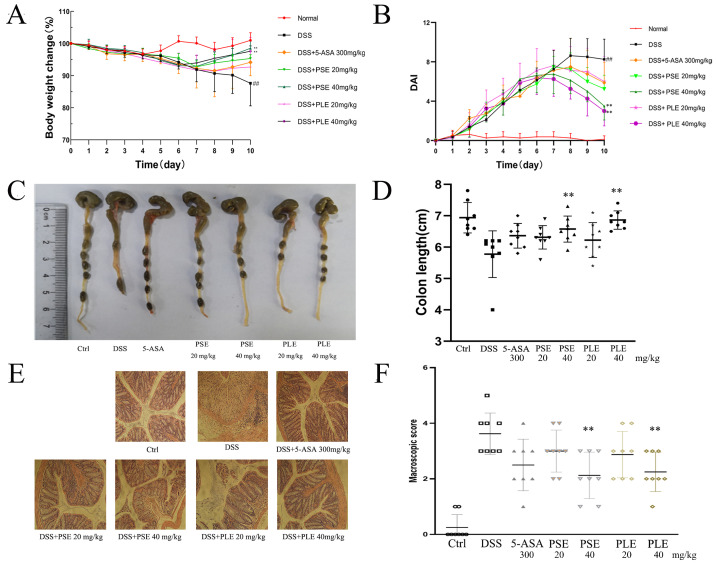
PSE and PLE ameliorated DSS-induced colitis in mice. Mice were administered 3.5% dextran sulfate sodium (DSS) (day 1 to day 7) in drinking water with or without PSE and PLE (20 or 40 mg per kg, day 3 to day 10). (**A**) Changes in body weight detected during disease progression. (**B**) Changes in disease activity index (DAI) levels were evaluated every day through all 10 days of the experimental period. (**C**) The colon length observed on the end point of the experiment. (**D**) Statistical analysis of colon length in different groups. (**E**) HE staining of colon tissue sections (original magnification 200×). (**F**) Histological score of the colon in each group. Data represented as mean ± SD of three experiments. * *p* < 0.05, ** *p* < 0.01 vs. model group (DSS).

**Figure 3 antioxidants-10-01684-f003:**
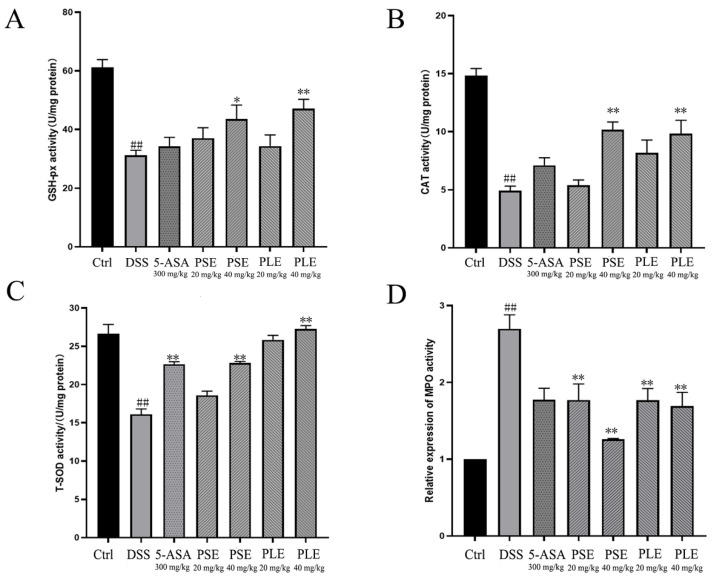
Effects of PSE and PLE on the activities of oxidation related enzymes in colon tissues of mice with DSS-induced colitis. Different groups of colonic sites were lysed using lysates and the glutathione peroxidase (GSH-Px) (**A**), catalase (CAT) (**B**), total superoxide dismutase (T-SOD) (**C**), myeloperoxidase (MPO) (**D**) activities in the colon tissues were measured. Data are represented as mean ± SD of three experiments. ^##^
*p *< 0.01 vs. Control (Ctrl), * *p* < 0.05, ** *p* < 0.01 vs. model group (DSS).

**Figure 4 antioxidants-10-01684-f004:**
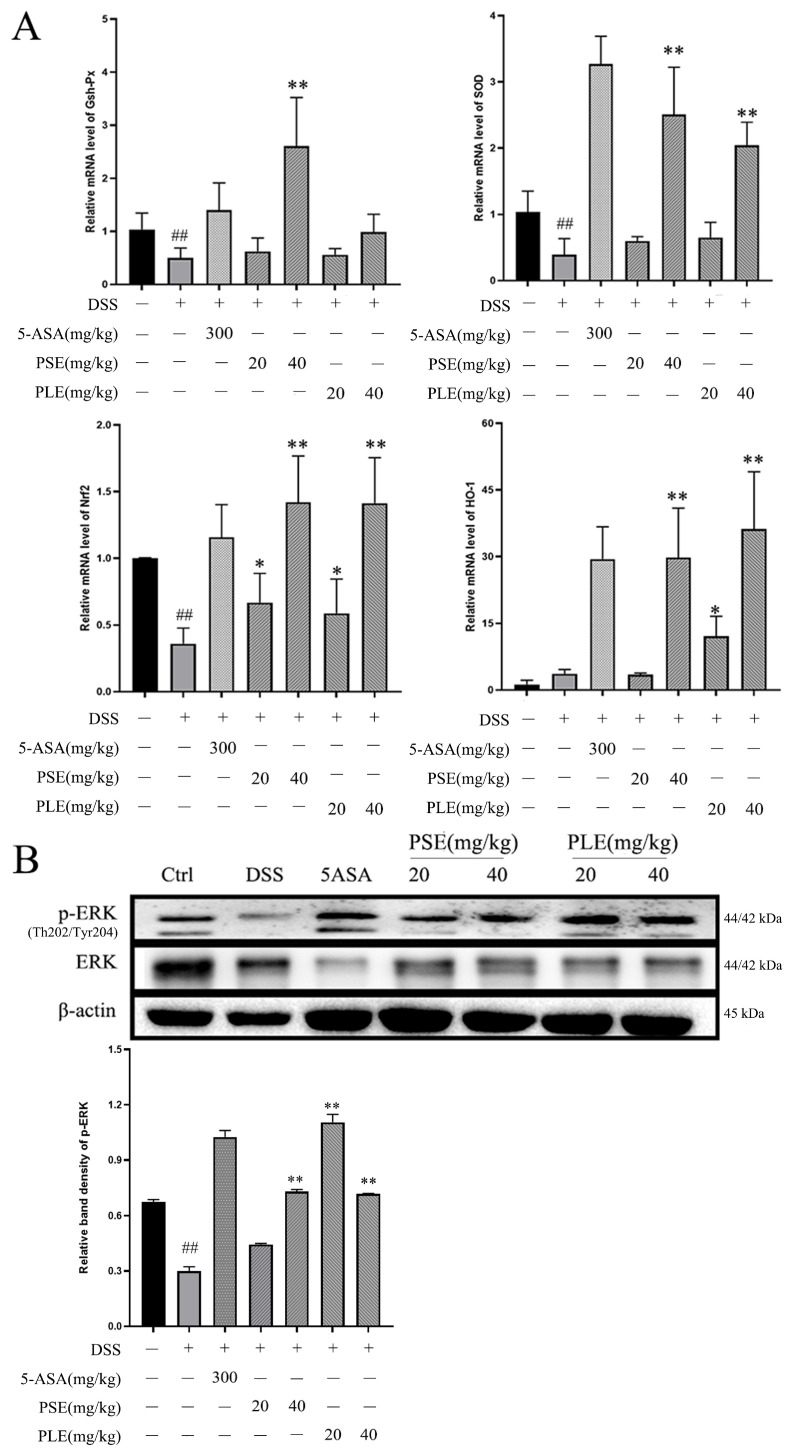
PSE and PLE suppressed oxidative stress by activating ERK/Nrf2 in colon of DSS induced colitis model. (**A**) The mRNA levels of *gsh-px, sod, nrf2 and ho-1* were measured by Q-PCR analysis. (**B**) The protein levels of p-ERK and ERK in the colon tissue were measured by western blot analysis. Data are represented as mean ± SD of three independent experiments. The “+” indicates that the substance (3.5% DSS) was added, and the “–” indicates that the substance was not added. *^##^ p <* 0.01 vs. Control (Ctrl), ** p <* 0.05, *** p* < 0.01 vs. model group (DSS).

**Figure 5 antioxidants-10-01684-f005:**
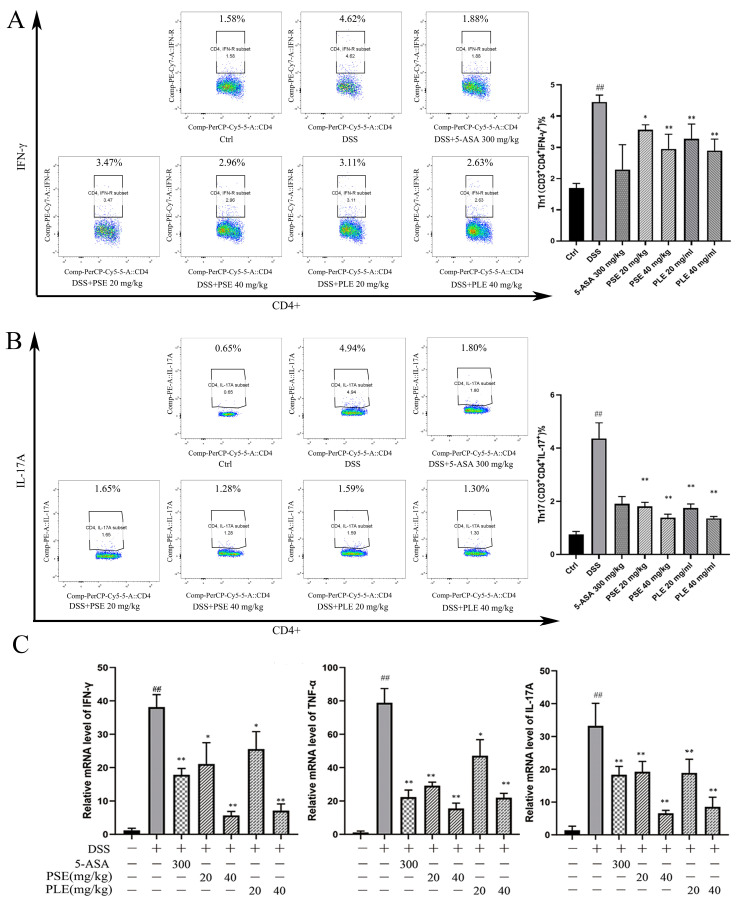
The effect of PSE and PLE on the expression of inflammatory factors in the spleen of mice with colitis. Splenic lymphocytes were obtained on day 10 and stimulated by PMA and BFA for 4 h. The intracellular IFN-γ (**A**) and IL-17A (**B**) production in CD4^+^ T cells were measured by flow cytometry. (**C**) The mRNA levels of *ifn**-γ, tnf**-α* and *il**-17A* in the spleen were assessed by quantitative RT-PCR. Data are represented as mean ± SD of three independent experiments. The “+” indicates that the substance (3.5% DSS) was added, and the “–” indicates that the substance was not added. *^##^ p* < 0.01 vs vehicle group, * *p* < 0.05, ** *p* < 0.01 vs. model group.

**Figure 6 antioxidants-10-01684-f006:**
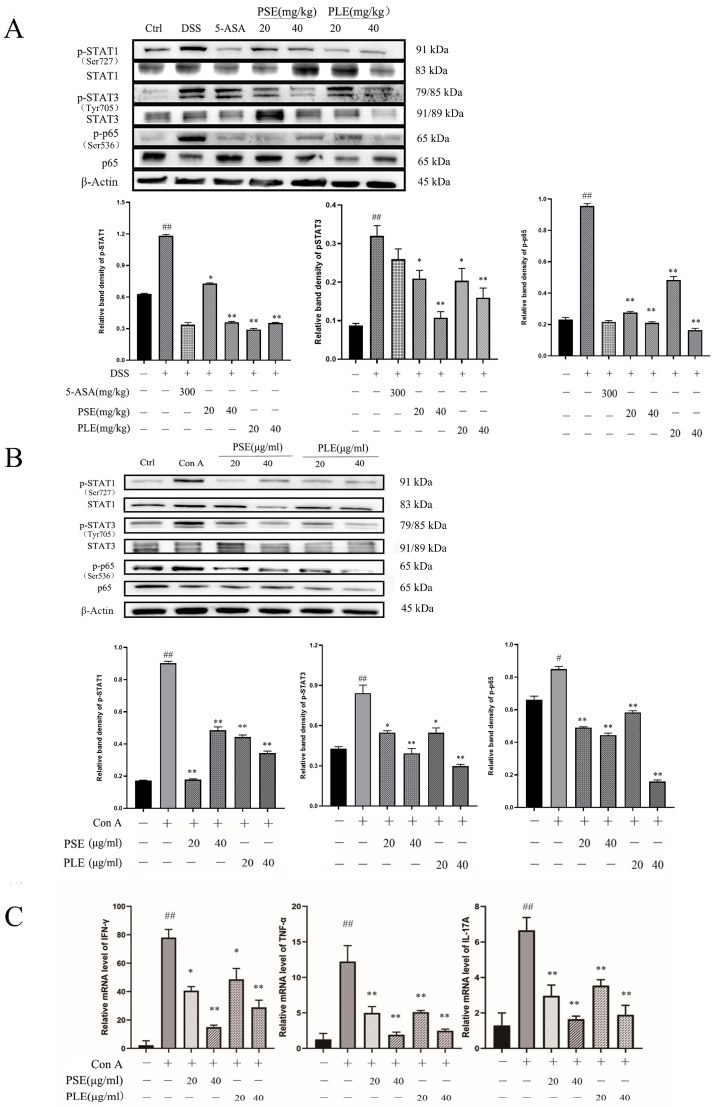
PSE and PLE inhibited phosphorylation of STAT1, STAT3, and p65 in DSS-induced colitis mice and Con A-activated lymphocytes models. Proteins were extracted from the spleen tissue and lymphocytes. (**A**) Expression of p-STAT1, p-STAT3, and p-p65 in spleen proteins of colitis mice was detected by Western blot analysis. (**B**) The expression of p-STAT1, p-STAT3, and p-p65 in lymphocytes were detected by Western blot. Band density of each protein was determined by Image J and normalized to the corresponding band density of the control protein (Actin). (**C**) The mRNA levels of IFN-γ, TNF-α and IL-17A in the lymphocyte were assessed by quantitative RT-PCR. Data are represented as mean ± SD of three experiments. The “+” indicates that the substance (3.5% DSS or 4 μg/mL Con A) was added, and the “–” indicates that the substance was not added. ^##^
*p* < 0.01 vs. vehicle group, * *p* < 0.05, ** *p* < 0.01 vs. model group.

**Figure 7 antioxidants-10-01684-f007:**
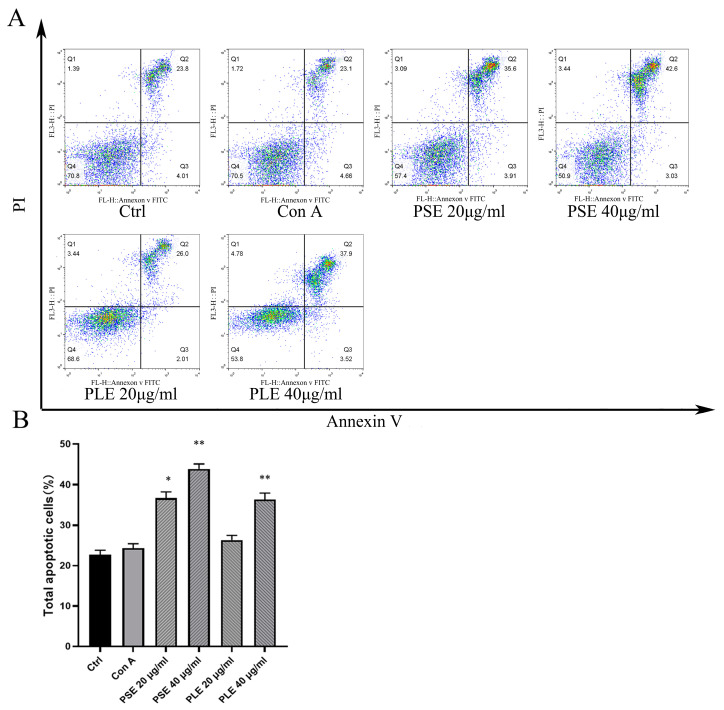
PSE and PLE can induce the apoptosis of Con A-activated lymphocytes. (**A**) Lymphocytes obtained from BALB/c mice were incubated with 4 μg/mL Con A and different concentrations of PSE and PLE at 37 °C for 24 h. Cells were stained with Annexin V and PI, and the cell apoptosis was measured by flow cytometry. (**B**) Annexin V-positive cells were analyzed as total apoptotic cells by column chart. Data are represented as mean ± SD of three independent experiments. * *p* < 0.05, ** *p* < 0.01 vs. model group.

**Figure 8 antioxidants-10-01684-f008:**
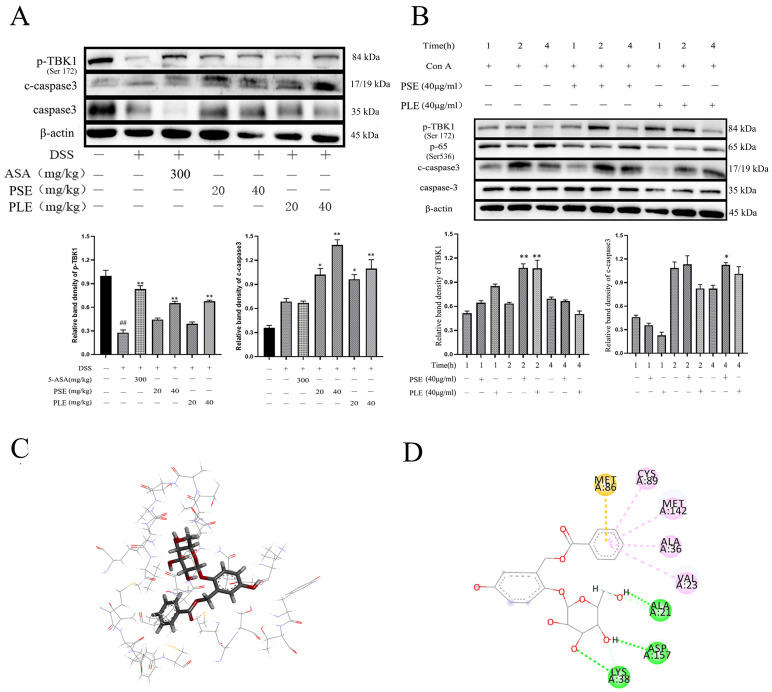
PSE and PLE may induce the apoptosis of lymphocytes through STING-TBK1 pathway. (**A**) The protein levels of p-TBK1 and c-Caspase3 in the spleen tissue of DSS-induced IBD mice were examined by Western blot. Band intensities of Western blots were normalized to actin and quantified by densitometry. (**B**) Lymphocytes were incubated with 4 μg/mL Con A and different concentrations of PSE and PLE at 37 °C for 1 h, 2 h or 4 h. The protein levels of p-TBK1, NF-κB and c-Caspase 3 were detected by Western blot. Band intensities from Western blot were normalized to actin and quantified by densitometry. (**C**,**D**) Induced-fit docking analysis of the interaction between salireposide and TBK1 protein. Hydrogen bonds and π-π stacks are indicated by green and pink dotted lines, respectively. The “+” indicates that the substance (3.5% DSS or 4 μg/mL Con A) was added, and the “–” indicates that the substance was not added. ^##^
*p* < 0.01 vs. Control (Ctrl), * *p* < 0.05, ** *p* < 0.01 vs. model group.

**Table 1 antioxidants-10-01684-t001:** Primers for Q-PCR.

Gene Forward	Gene Accession Number	Reverse
***nrf2*** AACAGAACGGCCCTAAAGCA	AH006764	TGGGATTCACGCATAGGAGC
***ho-1*** CACGCATATACCCGCTACCT	NM_010442	CCAGAGTGTTCATTCGAGCA
***sOd-1*** GTGTCTGTGGGAGTCCAAGG	NM_011434	CCCCAGTCATAGTGCTGCAA
***gsh-px*** ACAGTCCACCGTGTATGCCTTC	NM_001329527	CTCTTCATTCTTGCCATTCTCCTG
***tnf-α*** TGAACTCGGGGTGATCGGTC	NM_001278601	AGCCTTGTCCCTTGAAGAGAAC
***inf-γ*** TGAGTATTGCAAGTTTGAGGTCA	NM_008337	CGGCAACAGCTGGTGGAC
***il-17a*** TCGAGAAGATGCTGGTGGGT	NM_010552	CTCTGTTTAGGCTGCCTGGC
***β-actin*** TGCTGTCCCTGTATGCCTCT	AY618569	TTTGATGTCACGCACGATTT

## Data Availability

The data presented in this study are available in article.

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
