# Peer review of "Extracts of Waste from Poplar Wood Processing Alleviate Experimental Dextran Sulfate-Induced Colitis by Ameliorating Oxidative Stress, Inhibiting the Th1/Th17 Response and Inducing Apoptosis in Inflammatory Lymphocytes"

_antioxidants, 2021, doi:10.3390/antiox10111684_

Round 1

Reviewer 1 Report

The manuscript 'Extracts of waste from poplar wood processing alleviate experimental dextran sulphate-induced colitis by ameliorating oxidative stress, inhibiting the Th1/Th17 response and inducing apoptosis in inflammatory lymphocytes' is clearly described and rich in content.
The Introduction section frames the general topic of the study and highlights the reasons for this research.
The materials and methods are clearly and completely described with the appropriate bibliographical references.
The results are presented in a logical sequence in the text and respect the objective of the work. 
The discussions are well described and supported by bibliographical studies.

Reviewer 2 Report

In this submission, the active ingredients in poplar sawdust (PS) and poplar leaves (PL) were extracted and isolated, and their anti-inflammatory effects on mice with dextran sulfate sodium (DSS)-induced colitis were investigated. I like to give the following comments.

  1. Poplar sawdust and leaves need the authentication by expert.
  2. The positive control 5-ASA (300 mg/kg) reminded obscure. Please explain in clear.
  3. Mice fed with 3.5 % (wt/vol) DSS dissolved in drinking water for 7 days was also unknown. They drink the water in free? For 7 days needs the reference(s).
  4. Success of model induction was not described in detail.
  5. Western blots must show the antibodies in clear and followed the references.
  6. Contents of each phenolic glycoside were not quantified. Why?
  7. MPO levels increased in DSS-251 treated mice were inhibited by PME and PLE. However, the enzymatic activities of GSH-Px, CAT and T-SOD were not changed in parallel. Why?
  8. In Figure 4, unit of each signal in PCR must show in clear. Quantification of data in Western blots must indicate in detail.
  9. In Figure 6, data of Western blots must revise carefully. Activation of each signal such as STAT1 shall be calculated according to references.
  10. Docking analysis of the interaction between salireposide and TBK1 protein was not included in the methods.
  11. Lymphocyte apoptosis induced by PSE and PLE seems associated the STING/TBK1 pathway that needs more evidence.
  12. In conclusion, PSE and PLE possess anti-inflammatory and antioxidant activities that may be used to alleviate DSS-induced colitis in the future. Role of salireposide was ignored. Why?

Round 2

Reviewer 2 Report

It has been revised according to comments.
